# Religious Freedom, Cybersecurity, and the Stability of Society: Problems and Perspectives from a European Perspective

**Piotr Roszak** [1,*] and **Sasa Horvat** [2]

1 Faculty of Theology, Nicolaus Copernicus University, 87-100 Toruń, Poland
2 Faculty of Medicine, University of Rijeka, 51000 Rijeka, Croatia; sasa.horvat@medri.uniri.hr
* Correspondence: piotrroszak@umk.pl

**Abstract:** Although religious freedom significantly affects certain people, the guarantees for its observance also have implications for the quality of social life and the security of the state. Polarization and conflict between religious groups is not only a conflict for new believers, but also contributes to the weakening of the internal state. It seems that one of the elements of such a destabilization of states is the promotion and lack of reaction to the phenomenon of ridiculing religion and its followers in cyberspace. As can be seen from reports on the situation in Poland, there are increasing signs of hostility to religion, stereotyping, and discrimination. The issues can be considered on two levels: individual protection for freedom of speech, and state protection. At the end of this paper, proposals for measures to prevent crimes based on religion or belief are presented.

**Keywords:** religious freedom; cybersecurity; online religion; religion and security; freedom of religion

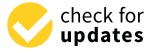



## 1. Introduction

The reports of many international organizations (including the OSCE for 2019–2020) highlight the importance of cyberspace for the development of individual and collective freedoms, and among them, there is one that is mentioned almost above all others and that belongs to the so-called first generation of human rights. It is religious freedom, a right that defines a person's identity in a transcendent dimension, including the forum of conscience, and that affects the perception of oneself in social reality (Huzarek 2021). The European Convention on Human Rights (1953) understands freedom of thought, conscience, and religion as a fundamental human right (Article 9). Further on, Article 14 of the Convention prohibits discrimination on the basis of religion, among other possible grounds (Horvat 2019, p. 7).

In the "classical catalogue" of the most typical offences against religious freedom, the most common are physical restrictions on the possibility of public worship; with the emergence of electronic media, however, new forms of discrimination against people on the basis of their religion and belief have emerged. Their origin lies not only in interfaith relations or in the radicalization of certain intra-religious groups, but can often be exploited by other forces seeking to pursue a policy of destabilization based on religious divisions (Tesler et al. 2019).

This situation is related to the cybersecurity of the state, for which the protection of religious freedom is of strategic importance. It is related to the concept of personal security, which is the ability to harmonize the self-determination of a person who shapes himself with the social, organizational, and technological infrastructure. The degree of harmonization depends on many cultural factors, primarily related to the understanding of what it means to be a human being, and thus to anthropology. According to Grabińska (Grabińska 2019), this ontic-cultural layer (1) is important for the perception of oneself and one's social role, that is, another psycho-social layer (2) of security. This layer, which shows courage or resistance, guarantees the cohesion of the community, based on the affirmation

of the other, and leads to the instrumental-defensive layer (3), thanks to which the person is able to responsibly protect himself and others by all means. The way to this synchronization is through education, which makes it possible to create appropriate security structures, and it is linked to the cybersphere, which can carry out actions that destabilize the process.

From the perspective of state interests, the protection of religious freedom is crucial for social security, as it guarantees the coexistence of people and strong social cohesion. At the same time, religious freedom preserves the transcendent dimension of human life, which makes it possible to overcome one's weaknesses: instead of submitting to technology (such technification can lead to the weakening of security), it preserves for people a sense of self-subjectivity that enables them to overcome their difficulties. To strengthen this capacity, it is especially important today to work on cyber-resilience. Failure to address the relationships between religions and attempts to distort them have concrete security implications. Intentional confusion in this dimension may be part of a broader strategy to destabilize the political community.

Unsurprisingly, religious freedom is relevant to securitization theory, in which otherness and difference play a key role and are embedded in broader contexts. At the same time, it is not enough to follow general trends (for example, regarding religious conflict in the world); security is strengthened by considering local contexts:

> "we need to remain open to what is happening in the digital lives of others, to the new vulnerabilities they confront, to the new forms of control that are deployed against them, to examine the trends in territories that might get ignored in our (imaginary) maps of digital geopolitics. But in the process of 'globalizing' these research questions, we should not lose sight of the need to think about the organized irresponsibility that we might find in the organizations and institutions that pride themselves as being the most 'advanced', 'rational' and 'cutting edge'." (Lacy and Prince 2018)

This article uses data from recent reports to analyze the most common forms of attacks on religious freedom in cyberspace (ranging from restrictions to ridicule of religiosity), followed by a discussion of the importance of this freedom from the point of view of state interests and an analysis of the possible scopes of state activity for which the protection of religious freedom in cyberspace remains important for security. This means activities in the area of preventing the ghettoization of religion and supporting de-radicalization processes.

## 2. Terminological Clarifications

In this section, we would like to give the reader a brief insight into the definitions of certain concepts we use, which at the same time offers a kind of theoretical "state of the art" of the topic we are dealing with.

Religion and religious behavior are present in all known human cultures; therefore, it is not easy to define it. However, we can state that religion is the relationship of man with the sacred, "whereby 'relationship' can have a great breadth of meaning and include theoretical, aesthetic and ethical religious act" (Rahner and Vorgrimler 1992, pp. 486–88; Donati 2019). St. Thomas Aquinas puts it nicely that all those who honor and serve God can be called religious.

As with all areas of human lives, religions are also deeply affected by new technological advances, which reshape religious organizations and practices of the faithful, especially the way religious beliefs are spread and communicated (Obadia 2017). The Internet is increasingly integrating into the religious dimension of human lives (Jonveaux 2021, p. 69). How much the Internet shapes and how much it is present in people's lives, affecting their religious life, is also shown by the fact that abstinence from use of the Internet is an increasingly common religious practice during "strong" religious holidays (for instance, during Lent for Christians).

Life in cyberspace is considered almost identical to real life, and this is also a fact when it comes to religions and religiosity on the Internet. Events in the space of the Internet are

in some ways a mirror image of what happens in real life. A large part of the religious life and rituals is also reproduced in cyberspace.

When we talk about religion on the Internet, the division into "religion online" and "online religion" is accepted. The first refers to the transfer of information related to the practices and activities of certain religious groups (for example, current events in a particular parish). The second term refers to the invitation to believers to participate in online religious rituals, for example, the transmission of Holy Mass over the Internet (Dawson and Cowan 2004, pp. 6–7). For Gasser, online religious communities are real communities because individuals participate in online religious rites and understand themselves as an integral part of that event and community; in this way, online activity directly shapes their real life (Gasser 2021, p. 183).

As for the concept of cybersecurity (or cyber security), it has its own history of development, and a common definition has not been agreed upon at the global level. On the contrary, the definition depends on the perspective from which cybersecurity is considered—industrial, state, or academic. The concept is developed and operates as a political and social construction (Kerttunen and Tikk 2020, p. 1), with more than 400 unique definitions (Deibert 2018). After analyzing numerous definitions, Schatz et al. singled out the key contents relevant to the definition of cybersecurity, which we transfer here in their entirety: "The approach and actions associated with security risk management processes followed by organizations and states to protect confidentiality, integrity and availability of data and assets used in cyber space. The concept includes guidelines, policies and collections of safeguards, technologies, tools and training to provide the best protection for the state of the cyber environment and its users" (Schatz et al. 2017, p. 66). Such a broad definition includes the protection of Internet users as well as their human rights, which in turn includes the fundamental right of religious freedom. In this way, the aim is to reconnect technologies to values, especially human rights (Garcia-Segura 2020, p. 39). A clear-cut overview of the issues can be found in The Charter of Human Rights and Principles for the Internet, which has an aim "to provide a recognizable framework anchored in international human rights for upholding and advancing human rights for the online environment". It is also interesting that the Charter emphasizes how "human rights apply online as they do offline".

In this regard, it is also worth mentioning the relatively new concept of the "cybersecurity culture", which stresses the importance of understanding a specific culture prior to an attempt to change or improve cybersecurity in that culture. This is because culture shapes the worldviews of individuals and influences how they react, behave, and judge in the real world, but also in cyberspace (Reegård et al. 2019, p. 4037). One of most fundamental factors of any culture is religion. This is another argument in favor of taking religion as a significant factor in cybersecurity issues.

It is not surprising that human rights have become debated issues in relation to new technologies. Kerttunen and Tikk state that some countries emphasize the freedom of expression and free flow of information, while other countries stress the challenges to sovereignty that usage of the new technologies may pose. Therefore, Kerttunen and Tikk claim that "how to protect universal and individual human rights is a core question within this often-binary debate" (Kerttunen and Tikk 2020, p. 2). It is also important to put forward the opinion that cybersecurity can affect international peace and security, while stability in cyberspace can contribute to international peace (Kerttunen and Tikk 2020, p. 5).

Ordinary people, on the other hand, understand by the term "cybersecurity" the ability to operate freely through the network without compromising their property, privacy, or personal rights (Kulesza 2016, p. 1). Nevertheless, the Internet has provided a space for the development of antagonisms and anti-religious movements (Dawson and Cowan 2004, p. 7).

## 3. Forms of Attacks on Religious Freedom in Cyberspace

The level of security is not only influenced by the structural activity of state organisms, but it is worth complementing it with a human-centered approach, where access to unbiased information also counts as a security factor that enables a free world of thought (Deibert 2018, p. 422). This ecosystem or information environment also applies to the topic of religion, and the concrete protection of personal rights depends on it: to limit expression is to limit progress, to nurture the message is to remove barriers. This fits with the right to privacy (Taddeo 2013; Hatzismay 2021), the tracking of which can limit the security of the individual—taking into account private activities to assess creditworthiness, for example, and more recently in China, tracking the activities of believers through special devices installed in temples or through apps. That is why today we are talking about data stewardship and emphasizing the need to protect people's information in cyberspace. (Floridi and Taddeo 2016).

### 3.1. Limiting the Activity of Religious Communities: Harmful Algorithms, Deletion of Religious Contents, and So On

According to the reported cases, the restriction of religious freedom in cyberspace can be both overt and covert, depending on how the intentions are formulated. In the political system of some countries, there are explicit provisions denying the possibility of religious communities and individual believers to express their views, and sometimes, with stated openness, there are instruments that effectively prevent the exercise of this right.

On the one hand, religious content is indirectly prohibited by preventing any material or press coverage of religious events. This can result from either the adoption of an official state religion or a strong version of secularism (Lazaro Pulido and Anchustegui Igartua 2021). In Western countries, this is the case that determines the exclusion of any reference from the digital space—an example of this is the behavior of the media, which, for example, eliminates names with religious connotations, such as "Christmas", or the mention of Christian traditions in advertising. This happened, for example, in 2020, with France's Inter radio, which initially refused to air a spot commissioned by "Oeuvre d'Orient" that drew attention to religious persecution in the Middle East because the word "Christians" appeared. However, after several days of public debate, Inter agreed to air the spot. The result is the creation of an alternative, unreal world based on an information bubble, and thus the creation of a reality based on criteria that are arbitrarily decided. Religious groups are locked in their information ghettos and treated as non-existent in social reality.

Banning religious content in cyberspace takes many forms. As reported by the Religious Freedom Lab, on 1/17 April 2020, an anonymous hacker disrupted the Internet broadcast of the Holy Mass from the Church of St. Elisabeth in Jaworzno, which was made available on its YouTube channel. This type of disruption of religious practices during a pandemic exposed a dangerous vulnerability in the security structure: the ability to cut citizens off from their connection to the community, but also to manipulate their messages, which could have implications for defending against an aggressor in the event of a conflict. The potential for this type of tainted messaging can be evidenced by other events, which, although they had the nature of individual occurrences, were a form of building tension between the religious and the rest. In the context of questioning the authority and achievements of St. John Paul II, a group of Internet users changed the names of streets and monuments dedicated to St. John Paul II on Google Maps.

Another case is the control of search engines/browsers in such a way that content is practically unavailable, even though it is officially claimed to be within reach of the audience. On the one hand, one builds situations where groups with a similar profile get the content they want, and so missionary activity suffers; because they never get offered other challenges, they are effectively shielded from this. This narrows the horizon and builds a world "on a small scale," which is dangerous for the state because it encloses citizens in capsules, so to speak. Preferences are an indicator that facilitates exploration, but they can also become a tool for building a vision of the world and segregating the

content that reaches people. It creates an illusion regarding the marginal importance of religion or even leads to the consolidation of a social (and legally supported) hostile attitude towards religion, while it remains an important factor that determines a person's identity (Kudła 2018).

### 3.2. Discrimination against Religious People for Their Beliefs on the Web

As Puppinck observed, religious freedom is a very broad and complex concept (Puppinck 2020). It refers to the freedom of manifestation of religion and not being constrained to practice religion, in particular through the teaching and practice of rites. There is a certain ambivalence in this notion, for since antiquity it has meant the right to believe and not to believe, to practice and not to practice, but the negative side of this right does not lead to the elimination of beliefs of other people. This means that the presence of religious symbols in public spaces cannot be challenged by the disbelief of others. Alejandra Vanney refers to this two-directed aspect, claiming that religious freedom as a guarantee of non-constraint to act according to one's own conscience should be combined with the duty of others to refrain from coercive action in matters of conscience (Vanney 2020, p. 22). Religious freedom is related to freedom of conscience, which protects the unity of life (i.e., to practice some professions and be faithful to religious convictions).

The Internet is a source of information about religious doctrines and the activities of individual communities, making it a key means of guaranteeing religious freedom, which from the beginning was understood as the freedom to worship externally, communally, and individually. As the reports show, this freedom of expression and publication, fundamental to freedom of religion, carries with it other phenomena based on knowledge of religiosity or lack thereof in specific citizens or communities. A study conducted by IWS in Poland in 2019 on a representative sample showed that disclosure of religiosity often leads to treating such people as inferior and evaluating the opinion of the (religious or non-religious) interviewee as worse (Roszak et al. 2022). Given the overwhelming religious majority in Polish society, these situations—referred to as microaggressions (Hodge 2019)—mainly involve religious people, and the means of expressing this hostility is cyberspace.

Information on religiosity obtained through cyberspace, as shown by the Laboratory of Religious Freedom in Poland report (LWR 2020, p. 271), translates into discriminatory behavior against people who declare themselves to be believers or not. An example would be the denial of access to certain services in the fitness industry to a person who appeared in social media to be critical of the attacks on Christian shrines and religious symbols that occurred during the so-called women's strike after the verdict of the Constitutional Tribunal of Poland confirming the right to life.

In this category of actions, it is worth mentioning articles or publications with ideological overtones written against religious people and religious teaching in schools especially, which are particularly present in Spanish electronic media. It should be emphasized that these are not opinions that are legitimate in a democratic society and related to the presence of religion in public space, but ideological calls to act against religious groups, to segregate, not to serve, and to deprive of rights (e.g., in education). Cyberspace has thus become a digital continent fertile for developing aggression towards religious groups.

Another example is from Croatia. Dissatisfied with the decisions of official bodies regarding measures to combat pandemics, which, among other things, allowed celebrated masses with a smaller number of people but also closed fitness centers, two men broke into a church in Zagreb, brought exercise equipment, and exercised by the altar. They also recorded this and later transmitted it over the Internet. After it gained the public's attention, one of the men apologized and deleted the video, but it had already gone viral and divided the public on these issues. There are also many other recent examples from Croatia where cyberspace was used to encourage intolerance and attacks against the Catholic Church and believers.

### 3.3. Hate towards Believers and Their Religions

However, it seems that the most frequent of the attacks on religious freedom is the promoted hate speech against adherents of various religions, which according to the UN intensified during the COVID-19 pandemic, when responsibility for the spread of the virus was attributed to religious minorities, among others. Many law regulations in recent years on hate speech (e.g., Racial and Religious Hatred Act 2006 adopted by the Parliament of the United Kingdom) aim at a broad protection of persons on the basis of, among other things, their religiosity, protecting them not only in clear situations of speech calling for aggression, but also misrecognition. In the latter case, the situation is one of judging a religious person by only selected characteristics and reducing all members of a group to a single category, thus oversimplifying the identity of the individual (Brown 2015, p. 173).

Forms of hate speech include humiliation, as well as defamation, which involves false statements regarding a religious group and characteristically moves to action or reaction at an emotional level (provoking a kind of fight response). In order for such procedures to work, hate speech wants to take advantage of the moment before rational thinking, often using so-called "cybercascades", which are created when an Internet user, based on contact with users (e.g., the religious), in influenced by certain false rumors or fake news. A number of conditions contribute to this state of affairs, such as reliance on information provided by others who cannot verify it and pressure to accept the opinion of a group due to a desire for recognition on their part (Brown 2015, p. 80; Sunstein 2007, pp. 83–91).

This kind of hate speech recently found its concrete expression on 24 October 2020: activists running the website www.strajkkobiet.pl (accessed on 14 June 2022) publicly called for hatred against the hierarchy of the Roman Catholic Church and for attacks on sacred objects in one of their posts. These were published as part of the "Word for Sunday" campaign, which was intended to ridicule/mock similar Catholic activities related to the experience of Sunday and Bible reflection. Under this banner, they called for disrupting worship services by standing with banners during the so-called common prayer. All these incentives for action originated in a campaign organized in cyberspace, and as the events showed, they put a halt to the liturgy, for example, in the Catholic Cathedral in Poznan (LWR 2020, p. 13), and caused disruptions in St. James Church in Torun (LWR 2020, p. 93). Cyberhate also moved, according to such texts, to insulting representatives of religious groups after the liturgy was over.

In a slightly different form, however, some feminist groups have also carried out a series of hate-speech-based campaigns in Spain against Christians and their presence in society. They took advantage of Women's Day on March 8 and encouraged people, through social media accounts, to attack temples, which actually happened. This did not only target Christians, but in Tarragona, for example, also Muslims. The texts against religion and its followers are extremely aggressive in the Spanish media, as indicated by the monitoring of Observatorio de la Libertad Religiosa in Madrid: they suggest a lack of the right to speech for Christians and their representatives and redefine manifestations of religious freedom (e.g., the right of parents to educate their children according to their beliefs or values) as a sign of intolerance. The artificially created "semantic confusion" and personal attacks on religious believers are often supported by the alleged right to blasphemy as a form of freedom of speech.

### 3.4. Reactions to Ridicule of Religion

Yet another form of attack on religious freedom is parodies of religious themes, and mockery of celebrations particular to religious groups. The disrespect for religious views is not humorous, but a form of soft-strike against content relevant to believers.

An example is the appearance of Holy Mass parodies in cyberspace, in which religious rituals are imitated in a deformed form (e.g., in terms of the dress of priests or altar boys, who wear items intended to arouse ridicule). The broadcasting of such videos on the web leads to social polarization and humiliation of those who practice religious rituals. The

motivation is thus characterized by hostility towards religion and a desire to marginalize it, which is a restriction of the exercise of the right to religious freedom.

## 4. Religious Freedom and Its Importance for Cybersecurity

The situations described in international reports in which religious freedom is violated in cyberspace (and briefly typified above) are not merely sporadic; they violate an important social fabric and lead to antagonism. Raising awareness of the importance of concern for religious freedom (i.e., enabling the free exercise of religion and protecting the exercise of this freedom) is in the interest of the state itself, whose level of security increases both individually and structurally (Hertzke 2012). It is worth considering "why" it is beneficial to include this component that is related to the scope of religious freedom in descriptions and analyses of the security of each state or society.

### 4.1. Everyone Quarreling with Everyone as a Method of Destabilization

Religious freedom is one that supports the freedom of individuals and groups to express their beliefs involving their worldview. At the same time, the media and cyberspace play an important role in building social connections, a sense of justice, and approaches to peace and response to armed conflict (Seiple et al. 2013).

The state's concern for religious freedom should be driven by the belief that inter-denominational conflicts, fueled externally, can lead to polarizations that are harmful to communities. Such antagonizing division of communities internally may be the first stage of a conflict that, by manipulating religious issues, could weaken the response in the face of an aggressor attack.

At the same time, it is worth noting that the prevention of such attacks can be used by states to restrict the activities of certain religious or social groups, as is being done in the Asian context. As in the case of many rights, a reference to other freedoms is needed, but also an indication of the scope of validity. The state, by its excessive interventionism, may create a "culture of fear", which does not allow the exercise of freedom of expression. Nevertheless, in the hierarchical arrangement of freedoms within the framework of human rights, religious freedom to practice one's beliefs is a first-generation right and the foundation of the others. It is in the interest of the state to ensure that there are no incitements to aggression against adherents of other religions, stereotyping, or verbal humiliation, which not only contradict human dignity but also lead to the de-stabilization of the state. A distinction must be made between criticizing the demands of a particular religious group and questioning its existence and social role. Inciting hatred between religions through the media may constitute the first stage of a hybrid war (Smuniewski 2019). By the same token, just as states are vigilant to ensure that calls for racial non-hatred (e.g., antisemitism) do not arise, so too should the scope of care address these fundamental issues rather than specific religious doctrines. The difference between criticism and insulting ridicule is increasingly well recognized.

It is important in this respect to examine the motivation behind attacks on religious freedom as such, but also the specific behaviors, especially in cyberspace, that limit the exercise of this right. As Puppinck suggests, this is related to a shift within social paradigms that involves the emergence of secularism as a belief in the existence of a neutral space free from the presence of religion. It is also related to the presentation of religion as harmful undertaken in modernism and sustained in the new atheism (Montoya 2022). As Blicharz emphasizes, these attacks often come from "anti-discriminatory positions regarding religion as a source of harm ( . . . ) The lack of balance between the value of antidiscrimination policy and the value of religious expression causes the diminution of the latter" (Blicharz et al. 2020, p. 413).

### 4.2. Religion as Protection of the Relations within the Society

The value of religious freedom for society is also evident in the fact that religious relationships contribute to the strengthening of social cohesion (Tole 1993). In the post-

Enlightenment tradition, religion was viewed as a mere set of beliefs (i.e., through the prism of ideology), whereas in the classical view, religion is a virtue that binds man to God (Strobel 2021). Religion is not about a certain canon of behaviors that is labelled as religious; everything that binds man to his ultimate goal can be regarded as religious. In this sense, religiousness encompasses man's entire life, so that every action can be religious insofar as it leads to the attainment of the ultimate goal (Roszak 2020).

Supporting religious freedom is therefore not a question of ideological preference, but a question of security, since it is the cementing of relationships within groups and ensuring their free functioning that holds the state together. This is confirmed by studies (Marcus and McCullough 2021) that show that socially desirable behaviors (such as lower rates of crime and delinquency) have much in common not so much with declared religiosity as with ritual practices (such as prayer). Thanks to religiousness, a person achieves self-control, so that in the event of conflict, he does not succumb to influences that would seek to take over the state.

*4.3. Respect towards Believers as Fundament of Stability and Social Peace (Identity Aspect)*

Another justification for the important role of the state in safeguarding religious freedom (Rawls 1958), contrary to the tendency to equate it with cultural rights or to treat it as a freedom far down the list of priorities, is to see in it the potential for social cohesion. Therefore, in considering the significance of religiosity for security, it is worth going beyond the secularist paradigm, which sees only threats in religious conflicts between different denominations or in religiously motivated fanaticism. Overlooking it is not the solution, but shaping it accordingly. Religion as such is not the problem, but the lack of opportunities for expression and unstable relationships, and especially the suppression of religiosity by secular ideology. Religiosity with its transcendent dimension is a proactive factor in creating stability and social peace (Puppinck 2020). This is due to the fact that religious groups are characterized by a strong identification and ability to defend values, including those related to social order. The challenge for security, therefore, seems to be to build a message of respect for adherents of different religions operating in the territory of a country, rather than treating them as a potential source of problems.

## 5. The Role of the State and Social Presence of Religion as Benefit

As we have seen, Internet content is playing an increasing role in the lives of citizens; at the same time, citizens are not well educated about all the possible dangers lurking in cyberspace. Therefore, continuous development of intervention programs is needed to improve prudent Internet skills (in Israel, as an example, it has become a social imperative) (Tesler et al. 2019).

Religion and the religious behavior of the human beings are important factors in cyberspace. Respecting and appreciating the dynamics that religion brings to a particular society (which we could see in part in previous sections), we can point out how important it is to protect spaces on the Internet for human religiosity.

In the context of the stability of the state, it is necessary to recognize the close connection between religions and societies, which also transfers over into cyberspace. In other words, cyberspace can be a factor that contributes to the development and refinement of that relationship, or it can be a "fertile ground" for disrupting that relationship and harassing believers, as we have seen in a few of these examples.

Although it has not yet been, and perhaps in the future will not be, strictly defined, cybersecurity should be a broad enough concept to include the dimension of religion. When states devise their cybersecurity strategies, it is fruitful to keep in mind the factor of religion.

From one side, studies have shown that religious people have different views on privacy, which in turn affect the way they use technology. Specifically, religious people are less inclined to share personal information over the Internet, which also points to religion as a positive factor in developing effective cybersecurity (Baazeem and Qaffas 2020, p. 111). In

this way, religion can contribute to global cybersecurity resiliency. In addition, theological studies, as scientific approaches to religion, can contribute to cybersecurity in matters of moral and ethical responsibility. It is about developing a "digital ethic" that looks at how we treat others in online environments, as well as what technologies and platforms are used to achieve it (Elia 2018; Beltramini 2021).

On the other side, cyberspace can become a space of ghettoization of religion and in this way contribute to instability in the real lives of citizens and society. Attempts at such "cyberspace ghettoization" have been observed in several cases we have cited, in which believers are told through cyberspace not to interfere in certain social realities.

Nevertheless, the positive impact of keeping religious communities connected through cyberspace during the pandemic was recognized in 2020 by the World Health Organization, which encouraged people to use technology to maintain community and continue worship.

## 6. Conclusions: The "Cyber-Culture of Transparency"

Understood in a wide arc, and bearing in mind above the human rights of individuals, cybersecurity should, in its perspective, aim at a culture of transparency that will contribute to the development of an open and inclusive society. This includes taking care to prevent the radicalization of cyberspace on the basis of religious divisions or worldviews.

Cyberspace thus becomes a new space of challenge for fundamental human rights and freedoms, such as the right to freedom of religion, but also the right to freedom of speech for believers. Here, we agree with Garcia-Segura that education and dialogue are two key elements for developing cybersecurity policies (Garcia-Segura 2020, p. 45). In the context of the topic of our paper, education can contribute toward better understanding of fundamental rights, including the right of freedom of religion. In this way, "educational measures that foster respect for religious or belief diversity are essential" (OSCE 2019, p. 20).

Further on, cyberspace offers a vast space of opportunities for the development of dialogue, which will enable a better acquaintance of all members of society and their worldviews and thus contribute to the stability and preservation of social peace. Here, we also stress, along with OSCE, that it needs to be ensured "that channels of communication are always open and that dialogue and engagement are developed on a regular and not a sporadic basis" (OSCE 2019, p. 23).

Cyberspace thus reveals itself to us as a new field that enables a deepening of the understanding of fundamental human rights, which then inevitably includes the issue of religions, as a recognized factor that contributes to the stability and security of society.

In this article, we have also touched upon the issue of the line between two human rights—the right to religious freedom and the right to freedom of speech. In the name of the right to religious freedom, we do not advocate censorship in the strict sense of the word. However, we hold that there is a need for a call for a stronger responsibility of European countries to recognize the existence and increasing occurrence of violations of the aforementioned right in cyberspace. Religious extremism has also found its place in cyberspace, and it also derives its certain impulse and strength from attacks on the human right to freedom of religion. Certainly, the best protection is the preventive education of members of all strata of society. However, in certain situations, the state should show stronger prevention.

**Author Contributions:** Conceptualization, P.R. and S.H.; methodology, P.R. and S.H.; writing—original draft preparation, S.H.; writing—review and editing, P.R. All authors have read and agreed to the published version of the manuscript.

**Funding:** This research received no external funding.

**Acknowledgments:** This research was possible thanks to the project Laboratory of Religious Freedom at Pro Futuro Theologiae Foundation in Poland. We are very grateful to all members of this team, in particular to Tomasz Huzarek, Weronika Kudła, and Alejandra Vanney.

**Conflicts of Interest:** The authors declare no conflict of interest.

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
