# Peer review of "Religious Freedom, Cybersecurity, and the Stability of Society: Problems and Perspectives from a European Perspective"

_religions, doi:10.3390/rel13060551_

Round 1

Reviewer 1 Report

No comments; article is excellent as is.

Author Response

Thank you for your opinion! 

Reviewer 2 Report

This manuscript advances a far-fetched interpretation of cybersecurity which, if it were to be applied as the Author expects, would render it possible to suppress all criticism of religion on the internet. It is clear that the promotion of hate speech and cyberbullying targeting faith communities should be actively combated by the state, but that is not the point, or in any case not the only point, of this manuscript. 

The Author freely mixes various types of threats perceived by religious people as targeted against the values they hold, these threats ranging from real and seriously disturbing, such as anonymous calls for organised attacks on sacred sites, to trivial and clearly exaggerated, such as ridiculing the religious robes worn by clergy of a particular tradition. The Author's apocalyptic tone is in stark contrast to the examples given in the manuscript: should a single hacking attack on a YouTube broadcast of a religious ceremony, or a training session held in a church by some frustrated subscribers of a gym which had closed due to Covid restrictions, really be discussed in terms of a major threat to religious freedom and national security? The writing style of this piece suggests that everything somehow connects to everything, from the minor annoyances which affect religious people on the internet to the physical attacks on temples in Spain. 

I would be prepared to accept the argument that stopping particularly malicious and brutal online attacks on religious people might be in the public interest. It probably is, because reducing religious tensions enhances social peace. But with the same in mind, restraining criticism of religion is definitely counterproductive for social peace. To sound more convincing, the Author would have had to at least outline the limits of acceptable criticism of religion and the limits of the State's responsibility for the behaviour of its citizens. There is a rich literature on this subject. After what I have read, I am concerned that the Author actually advocates censorship. And by framing the discussed issues as a matter of cybersecurity, the Author is apparently trying to pin the religious (and specifically Christian) agenda on the harshest legal measures used to combat terrorism and foreign infiltration. There are also some clear signs of bias here: the manuscript only discusses non-religious acts of hostility, while online religious extremism, which certainly has a negative impact on social peace and security, remains out of the spotlight, as the Author believes that such an approach belongs to the 'secularist paradigm' (354). 

Some terms used in the manuscript have been insufficiently explained or misunderstood. Since the Author refers to the legal prohibition of discrimination, one would expect this term to be used in its legal sense. If someone in Spain directly calls for religious people to be deprived of their rights, this should be the focus of the Author's concern, but mere 'articles or publications with ideological overtones written against religious people and religious teaching in schools' (232) certainly do not meet the criteria of discrimination. The manuscript also refers to 'microaggressions'—an ambiguous concept which has notably been popular among critical race theorists—but does so without attempting to explain it and ignores the extensive literature sceptical of the concept. 

For all these reasons, I cannot recommend this piece for publication. 

Author Response

We are very grateful to the Reviewer for his/her helpful comments. We tried to answer it in new version of the manuscript, at least to some of the Reviewer's suggestions. Nevertheless, it should be had in mind that there is a principal difference between our and Reviewer's opinion on religious freedom in Europe. Both are justified, but we prefer our perspective that we hope is well grounded in empirical research and literature. 

Unlike the reviewer, we do not consider it trivial to ridicule the religious robes worn by clergy of a particular tradition. We see the process of attack on religious freedom (and religion as such) more broadly, as beginning with stereotyping, through discrimination leading to persecution.

Because it is a scientific approach to the topic, we have tried to avoid an apocalyptic tone and only draw attention to the processes taking place in Europe (and different from those in Asia). Nevertheless, we have revised the style in this sense.

We draw attention to certain aspects of the phenomena that lead to restrictions on religious freedom. Data from many studies show a certain paradox related to the feeling of being attacked when manifesting religious views.

The concept of microaggression has not received much criticism, at least this is not reflected in the scientific literature. Besides, this connect of microaggression is used by often radically different groups, like LGBT and religious practitioners .  

As suggested, we have tried to clarify certain concepts used in the text.

We are aware that we will not take up all the Reviewer's suggestions, but our goal was to call attention to the perspective of cyberspace, where we observe some mechanism that limit religious freedom (to believe and not to believe!). Our role was humble and consist in mapping of the situation that can provoke more  researches in line with Reviewer's recommendation. 

Reviewer 3 Report

The manuscript examines the issues and perspectives about religious freedom, cybersecurity, and the stability of society. It discusses the attacks on religious freedom in cyberspace, emphasizes the importance of protecting religious freedom, and argues that it is the state’s interest to ensure religious freedom in cyberspace. The manuscript can be improved in the following ways:

  1. It is not clear what areas you focus on in your study. Are you discussing only the phenomenon that happened in Europe? If so, you need to mention that explicitly.
  2. You have discussed the forms of attacks on religious freedom in cyberspace. However, your discussion provides one dimension analysis. You need to discuss the perspectives of the people who attack religious freedom in cyberspace. In other words, what causes so much hatred against religious people in cyberspace.
  3. It would help if you were careful about your choice of words. In line 280, you talked about extreme feminist groups. It would be best if you defined what you meant by extreme feminist groups. In lines 353 and 354, you made a bold statement. Your statement blurs your position of neutrality as a researcher. You should discuss the secularist paradigm more broadly.
  4. In line 376, you may write “previous sections” instead of “previous chapters”.
  5. You used the term “religious freedom” many times in your article. You kind of explained what you meant by it in section 3.2. You may also discuss freedom of religion means freedom of observing religion or not observing religion.
  6. You may consider improving section 4 (Religious freedom and its importance for cybersecurity) by elaborating more. The subsections, for example, 4.1 is too short.
  7. Many countries promulgate laws against the acts of insulting religious sentiments in cyberspace. This practice creates a culture of fear. Sometimes governments exploit this law and use this law to silence detractors (example: Bangladesh). While discussing the states’ role in protecting religious freedom in cyberspace, you should discuss the loophole too.
  8. The languages of some paragraphs are ambiguous. It would be helpful to review your English language and style by professionals.
  9. Your conclusion ends abruptly. It will help if you write a proper conclusion that summarizes your findings.  

Author Response

 Thank you for all these helpful comments. We tried to implement ALL suggested improvements. Below is our response to each of the Reviewer's claims (in red). 

  1. It is not clear what areas you focus on in your study. Are you discussing only the phenomenon that happened in Europe? If so, you need to mention that explicitly.

Although our perspective concerns many cultural contexts, it is true that our main focus is applicable to the situation in Europe, so we will add this in the title in order to be clear. Thank you for helping us to be more precise in this point. 

2. You have discussed the forms of attacks on religious freedom in cyberspace. However, your discussion provides one dimension analysis. You need to discuss the perspectives of the people who attack religious freedom in cyberspace. In other words, what causes so much hatred against religious people in cyberspace.

The problem of “motivation” of people attacking religious freedom (RF) is surely another topic and we have investigated it broadly. We have preferred to focus on cyberspace and call attention to what is happening there, what kind of limitation of RF we can observe etc. But we will add a short description of it. Our diagnosis was suggested within the description of secularist paradigm as a main cause of observed attacks on RF, because of its claim that all expressions of religiosity should be eliminated from public space in order to avoid tension, conflicts within religious group etc.

3. It would help if you were careful about your choice of words. In line 280, you talked about extreme feminist groups. It would be best if you defined what you meant by extreme feminist groups. In lines 353 and 354, you made a bold statement. Your statement blurs your position of neutrality as a researcher. You should discuss the secularist paradigm more broadly.

Done. Some of the polemic indication was removed and added a short the explication why secularism consider the religion as source of harm with reference.

4. In line 376, you may write “previous sections” instead of “previous chapters”.

Done.

5. You used the term “religious freedom” many times in your article. You kind of explained what you meant by it in section 3.2. You may also discuss freedom of religion means freedom of observing religion or not observing religion. 

Done.

6. You may consider improving section 4 (Religious freedom and its importance for cybersecurity) by elaborating more. The subsections, for example, 4.1 is too short.

Done.

7. Many countries promulgate laws against the acts of insulting religious sentiments in cyberspace. This practice creates a culture of fear. Sometimes governments exploit this law and use this law to silence detractors (example: Bangladesh). While discussing the states’ role in protecting religious freedom in cyberspace, you should discuss the loophole too.

Done. Added to the section 4.

8. The languages of some paragraphs are ambiguous. It would be helpful to review your English language and style by professionals.

Done.

9. Your conclusion ends abruptly. It will help if you write a proper conclusion that summarizes your findings.  

Done.

Round 2

Reviewer 3 Report

The authors incorporated the reviewer's comments and improved the manuscript significantly.